# Primary care clinicians' views of paediatric respiratory infection surveillance information to inform clinical decision-making: a qualitative study

Emma C Anderson,[1] Joanna May Kesten,[2] Isabel Lane,[3] Alastair D Hay,[4] Timothy Moss,[5] Christie Cabral[6]

This research was presented at the General Practice Research on Infections Network (GRIN) conference, University of Oxford in September 2016, and was presented as part of the CAPC seminar series at the University of Bristol on 21 March 2017.

For numbered affiliations see end of article.

**Correspondence to**
Dr Emma C Anderson; emma. anderson@bristol.ac.uk

## ABSTRACT

**Aim** To investigate primary care clinicians' views of a prototype locally relevant, real-time viral surveillance system to assist diagnostic decision-making and antibiotic prescribing for paediatric respiratory tract infections (RTI). Clinicians' perspectives on the content, anticipated use and impact were explored to inform intervention development.

**Background** Children with RTIs are overprescribed antibiotics. Pressures on primary care and diagnostic uncertainty can lead to decisional biases towards prescribing. We hypothesise that real-time paediatric RTI surveillance data could reduce diagnostic uncertainty and help reduce unnecessary antibiotic prescribing.

**Methodology** Semistructured one-to-one interviews with 21 clinicians from a range of urban general practitioner surgeries explored the clinical context and views of the prototype system. Transcripts were analysed using thematic analysis.

**Results** Though clinicians self-identified as rational (not over)prescribers, cognitive biases influenced antibiotic prescribing decisions. Clinicians sought to avoid 'anticipated regret' around not prescribing for a child who then deteriorated. Clinicians were not aware of formal infection surveillance information sources (tending to assume many viruses are around), perceiving the information as novel and potentially useful. Perceptions of surveillance information as presented included: not relevant to decision-making/ management; useful to confirm decisions post hoc; and increasing risks of missing sick children. Clinicians expressed wariness of using population-level data to influence individual patient decision-making and expressed preference for threat (high-risk) information identified by surveillance, rather than reassuring information about viral RTIs.

**Conclusions** More work is needed to develop a surveillance intervention if it is to beneficially influence decision-making and antibiotic prescribing in primary care. Key challenges for developing interventions are how to address cognitive biases and how to communicate reassuring information to risk-oriented clinicians.

## INTRODUCTION

Antibiotics are overprescribed in primary care for respiratory tract infections (RTI),[1 2] contributing to antimicrobial resistance.[3 4] RTIs are the most commonly managed problem of childhood managed in primary care.[5] Diagnostic uncertainty is a key driver of antibiotic overprescription[6] with evidence of variation in prescribing between clinicians[7] and surgeries[8] potentially attributable to uncertainty regarding effective RTI diagnosis and treatment.[9] That non-clinical factors are known to impact prescribing variation accentuates this picture.[10–14]

### What is known about the subject?

► Children with respiratory infections are overprescribed antibiotics and we need interventions to aid decision-making and antimicrobial stewardship.
► The collection of real-time data on locally circulating viral infections is feasible and could be turned into an informational intervention to aid clinical decision-making.
► Parents are receptive to clinicians using online information of locally circulating viral infections within consultations.

### What this study adds?

► Primary care clinicians (general practitioners and nurses) self-identified as rational prescribers for paediatric respiratory tract infection (RTI), though cognitive biases in decision-making were evident.
► Responding to a prototype intervention of RTI infection surveillance, clinicians expressed wariness about using population data to influence their clinical judgement of individual children.
► Intervention development to aid primary care management of paediatric RTI needs to take careful account of clinicians' predominantly risk-oriented role.

**BMJ**

**Table 1** Cognitive biases in medical decision-making relevant to paediatric RTI

| Cognitive bias | Description | Example/consequence of relevance to paediatric RTI |
| --- | --- | --- |
| (1) Anticipated regret | The probability of a diagnosis with a severe outcome is overestimated due to a heightened sense of future regret in the event of missing the diagnosis. | Clinicians' fear of 'missing the sick child' leading to prescribing 'just in case',[41] due to perceptions that not prescribing carries greater potential threat. |
| (2) Anchoring and adjustment | Assessing new cases in relation to a previous case, rather than a population baseline. | Assessing a child's RTI as severe/not in comparison to the last sick child/ren seen, rather than as a new case against a broad population baseline. |
| (3) Confirmation bias | Selectively gathering and interpreting evidence to confirm a diagnosis, and ignoring evidence that may disconfirm it. | Deciding a child needs antibiotics based on a 'gut' feeling and looking for reasons to prescribe. |
| (4) The availability bias | Information that is easily recalled is given high importance. That is, salience correlates with decision-making, regardless of the quality of the evidence. Information salience is increased by being: frequent, recent, unusual, emotive or high profile. Research shows that simply *imagining* a diagnostic outcome (therefore making it salient) will raise a clinician's subjective probability of its likelihood.[42] | Remembering a child with RTI symptoms who deteriorated when not offered antibiotics; media reporting of a child deteriorating after seeing their GP. |
| (5) Representativeness | Assuming that what presents in clinic represents a 'real' state of events, includes: (A) not accounting for regression to the mean by assuming acute symptoms are representative of the illness, rather than an anomalous peak; (B) assessing only by the similarity of symptoms with possible diagnoses, and ignoring relevant base rate probabilities of diagnostic options; (C) the gambler's fallacy of reasoning that sequential cases represent the spectrum of probabilities, for example, after four similar successive cases given diagnosis A (80% probability), similar case number 5 is given diagnosis B (20% probability), rather than being assessed independently as having 80% probability of diagnosis A. | Prescribing antibiotics to a proportion of children presenting with RTI, based on symptoms on the day. |

GP, general practitioner; RTI, respiratory tract infection.

Primary care clinicians navigate uncertainty under pressured conditions (limited time, increasing complexity and workload)[15] that increase susceptibility to cognitive biases that influence decision-making[16 17] (table 1 shows relevant examples). Horwood and colleagues[18] show the dual processes of clinical decision-making (akin to Kahneman's system 1 and 2 thinking[19] in paediatric RTI, combining rapid 'gut feeling' (system 1) with detailed deductive reasoning (system 2)), and Djulbegovic and colleagues[20] show how these dual processes are linked with common cognitive biases.[20] Paediatric RTI management is characterised by fearing negative consequences, a cognitive bias of *anticipated regret*, leading to antibiotic prescribing 'just in case'.[14] Cultural roles of child as vulnerable and general practitioner (GP) as help-giver[14] add to this emotionally laden motivator, often exacerbated by salient media reports of negative outcomes for individual children. Antibiotics present an accessible risk management tool in this context.[14] Supporting deductive reasoning in this context may be helpful.

Variability in pretest probability estimates in clinicians[21 22] is thought to impact on diagnostic and treatment accuracy. Differences in subjective judgements of disease prevalence could account for some practice variation in prescribing rates.[20] Clinicians are recommended to begin their deductive reasoning process by consulting epidemiological sources, which need to be accurate and available.[22 23] Research calls for more detailed evidence around paediatric RTI prognosis to reduce uncertainty.[24] A systematic review showed that providing surveillance data to clinicians may have a role in reducing antibiotic prescribing in primary care, though the evidence was not robust, and the article highlighted the need for more research.[25]

Microbiological and syndromic incidence data are routinely collected (eg, by Public Health England and QRESEARCH). We have demonstrated the feasibility of primary care[26] and parent-collected[27] microbial sampling that could also be used for surveillance, the being the Evaluation of Enhanced Paediatric Respiratory Infection Surveillance (EEPRIS) study,[28] within which this qualitative study was nested. It is hypothesised that presenting local surveillance data online (in near real time) could provide relevant and accurate epidemiological baseline

information to aid probabilistic diagnostic reasoning. The ability to match patients' presentation with known circulating viral RTIs could reassure, reduce uncertainty around diagnosis and reduce prescribing bias.

Intervention development should follow an iterative programme of research and stakeholder contribution.[29–31] Interventions for antimicrobial stewardship in primary care paediatric RTI need to address clinician and parent needs, and involve GPs in its design.[32] Our parallel study showed parents were receptive to clinicians' use of surveillance information within consultation to support diagnostic and management decisions.[33] Gaining clinician views is key to effective intervention development.

## Aims
The aim of this study was to assess clinicians' perspectives on the EEPRIS surveillance information intervention, in order to inform its design (content and delivery).

## METHODS
### Patient and public involvement
A patient and public involvement (PPI) team of eight parents advised on all aspects of the EEPRIS study (from design to dissemination) within which this qualitative study was nested. They were consulted about the development of a prototype surveillance intervention, and advised on the design of two parallel projects involving parents.[33 34] Participants in the present study are clinicians, so we did not specifically consult our PPI group for clinician recruitment methods and interview conduct. Methods were based on our experience of interviewing GPs in previous research,[35] together with input from medical clinicians within the research team (ADH, IL) who advised on appropriateness of questions and participant burden. Once published, participants will be informed of the results through the EEPRIS website: http://www.bristol.ac.uk/primaryhealthcare/research-themes/eepris/study-outputs/

### Sampling
Eligible participants were GPs and nurse practitioners (NP) with paediatric prescribing responsibilities practising in surgeries in a South-West of England city. All eligible clinicians (n=89: 80 GPs, 9 NPs) identified by the practice manager at 10 surgeries participating in the EEPRIS feasibility study[28] were sent a participant information sheet describing the study via email. The aim was to recruit around 20 participants, representing a range of participant and practice demographics. Non-responding GPs and practices were contacted multiple times via email and telephone. Practice reimbursement of £40 was offered for the interview. Recruitment was ongoing throughout the data collection phase (February to July 2016) and ceased on reaching data saturation for core themes.

### Data collection
After written informed consent was recorded, semistructured interviews were conducted face to face with clinicians (GPs and NPs) within their respective surgeries, typically lasting 30 min. Interviews followed a topic guide which explored current approaches to managing paediatric RTIs and knowledge of circulating infections, then elicited views of a paper mock-up of RTI viral surveillance information, see figure 1. This included microbiological descriptors, syndromic details and a graph indicating recent prevalence of hypothetical 'top three viruses' in the local area, presented alongside typical symptom duration of common RTIs, taken from published research,[36] developed by IL (a medical clinician). Clinicians were encouraged to give uncensored (positive and negative) responses, in an aim to reduce the risk of tailoring responses towards perceptions of what the researcher may want to hear (known in psychological literature as 'demand characteristics').[37] Interviews were audio recorded, transcribed verbatim and imported into NVivo for analysis, ensuring secure (encrypted/password-protected) storage.

### Analysis
Inductive thematic analysis was applied,[38] comparing themes within and across the sample, structured around the interview topics. NVivo was used for coding the data to enable the inductive charting of themes by participant. Codes were assigned to the first few transcripts line by line to begin to summarise and interpret the data. Independent coding was conducted for accuracy checking (IL). Differences were resolved via discussion to ensure robust analysis. Codes were refined iteratively, condensing these into broader themes to produce an agreed set of codes to apply to subsequent transcripts, with regular meetings to reach consensus on coding and analysis.[39]

## RESULTS
### Sample description
Twenty-one clinicians (6 male, 15 female), consisting of 18 GPs and 3 NPs from eight urban GP surgeries were interviewed. GP surgery areas represented a range of neighbourhood deprivation levels, and clinicians represented a range of experience (1 to over 30 years practising) and a range of full-time and part-time working. Some had paediatric/respiratory interests or in promoting self-care; most had no special interest; others expressed topic relevance due to large volume of child consultations. Table 2 presents participant characteristics.

## ANALYSIS RESULTS
Results presented below are organised into (1) themes relating to existing practice and (2) themes relating to responses to the proposed intervention. Brief descriptions of each theme (shown in table 3) are presented

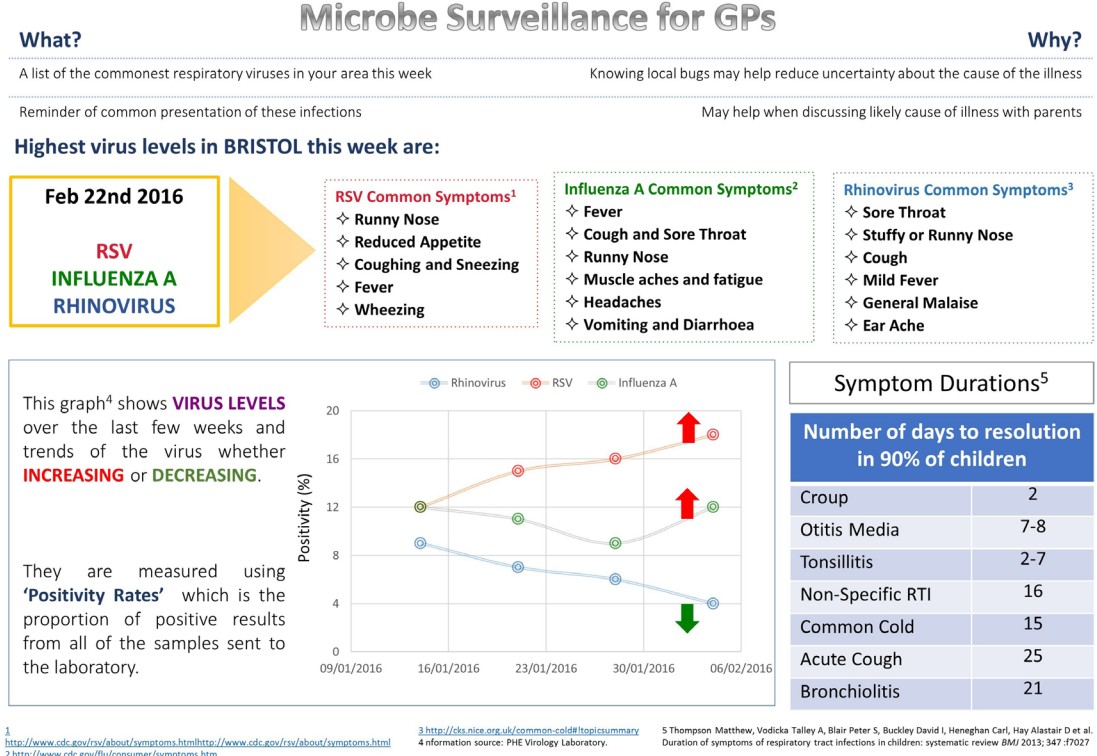

**Figure 1** Example RTI surveillance data. GP, general practitioner; RSV, respiratory syncytial virus; RTI, respiratory tract infection.

below, with key themes (highlighted bold in table 3) presented in more detail under respective subheadings.

### Existing practice

Diagnostic decision-making for paediatric RTI was characterised by uncertainty ('the smaller the kid, the harder it is to tell' GP3) and dual system processes ('[I] rely on observations; pulse oximetry, respiratory rate, pulse, and gut feeling to a degree' GP9). The role of the GP was described as to identify the serious rather than self-limiting illness ('so its self-limiting illness which, you know, they can manage at home vs serious illness that needs either antibiotics or hospital admission. That's essentially the GP job' GP5), or 'make that decision between viral or bacterial' (GP9) rather than specify between different viruses.

Clinicians presented their role as to treat each child as an individual, independent from the population, and in this context spoke of the need to make a full clinical assessment ('I just always assess them as new and they're all different, and they've got different histories and they've got different underlying illnesses, and they all just can respond differently' GP1). Clinicians described their perceptions of parents' worry, expectations for antibiotics and competence as influencing their prescribing decisions. No clear solution for addressing uncertainty was identified, with some citing point of care testing to aid prescribing decisions, several feeling happy with their own clinical judgement and current resources, others 'not sure' what could help.

### 'That's what really worries you with children': fear, risk and safety (children as a vulnerable group)

Questions about current clinical context elicited a combination of a fear of negative consequences, risk aversion and prioritising safety in decision-making. Worry was expressed about sick children in general, and respiratory illness in particular, heightened around younger children, represented as a particularly vulnerable group.

> That's the thing with children, isn't it? It's the respiratory failures that you worry… that's what really worries you with children. Anything that's kind of respiratory worries, me, because that's how children get really, really sick. (GP1)

### 'There's just lots of viruses in the winter': probabilistic reasoning

Most clinicians commented that they currently do not have, or know of, formal infection surveillance information, viewing such an intervention as novel and potentially useful. On exploring current means of gaining information on circulating infections, most clinicians cited what they see or hear in clinic ('the parents themselves tell you' GP5). No clinicians reported using existing epidemiological surveillance data for baseline probabilities of viral diagnosis or differentiating between viruses in decision-making. Most assumed general high viral prevalence.

## Table 2  Participant characteristics

| Participant | Gender (M/F) | Deprivation decile of practice area | Full-time (FT) or part-time (PT) working | Years practising (range categories) |
|---|---|---|---|---|
| 01 (GP) | F | 4* | PT | 10–14. |
| 02 (GP) | F | 4 | PT | 5–9. |
| 03 (GP) | M | 5 | FT | 5–9. |
| 04 (GP) | F | 6 | PT | 15–19† |
| 05 (GP) | M | 3 | PT/FT‡ | 0–4 |
| 06 (GP) | F | 3 | PT/FT‡ | 10–14. |
| 07 (GP) | F | 6 | PT | 5–9. |
| 08 (GP) | F | 8 | PT | 10–14. |
| 09 (GP) | M | 3 | FT | 15–19 |
| 10 (GP) | F | 8 | PT | 0–4 |
| 11 (GP) | M | 1 | FT | 10–14. |
| 12 (GP) | F | 1 | PT | 5–9. |
| 13 (GP) | F | 3 | PT | 30–39 |
| 14 (NP) | F | 5 | FT | 10–14. |
| 15 (GP) | M | 5 | PT | 20–29 |
| 16 (NP) | F | 2 | PT | 20–29 |
| 17 (NP) | F | 9 | PT | 20–29 |
| 18 (GP) | F | 5 | PT | 20–29 |
| 19 (GP) | F | 2 | PT/FT‡ | 0–4† |
| 20 (GP) | F | 9 | PT | 5–9. |
| 21 (GP) | M | 2 | FT | 0–4† |

*1=most deprived, 10=least deprived.
†Reported several more years practising as a doctor, before being GP.
‡Reported 'technically part time', though practising seven sessions a week.
GP, general practitioner; NP, nurse practitioner.

My perception is there's just lots of viruses in the winter and that's just how it is, and maybe it doesn't make that much difference what the viruses are. (GP12)

### 'If you've just seen a case… you are looking out for it': cognitive biases in decision-making

Despite cognitive biases being unconscious processes, interviews elicited some indication of 'anchoring and adjustment' (assessing cases against others seen rather than population baseline), 'availability' (decision-making influenced by salient emotive information) and 'representativeness' (not accounting for population base rate) biases in paediatric RTI decision-making (table 1). Some clinicians were aware of these processes. Two stated the impact of the nature of available salient information on increasing prescribing, while several described salient cases that impacted subsequent decision-making.

There is a risk that you're not gonna get that [reassurance] right every time, and if you see a thousand kids, you're gonna find one that actually did have a chest infection, and I probably see a thousand kids, I dunno, a year, maybe, and so I'm gonna get it wrong once a year. And then you're gonna have a kid that goes to A&E or goes to out-of-hours, and the parents think you're rubbish and all that kind of stuff, and you go, 'Well, I'll just treat an extra ten over-the-top,' and I think that's what the limiting factor is. (GP3)

If you've just seen a case of croup or err you've seen a hospital discharge talking about croup, then your antennae for croup is up undoubtedly, so you are then looking out for it. (GP11)

Anchoring and adjustment was shown mainly in the context of clinicians talking about their current (lack of) infection surveillance information, talking generally of assessing children against others seen in practice.

I guess we're not getting up to date microbiology advice, but you get a feel of what's out there, what type of symptoms children are struggling with. So I guess it's more anecdotal and what you're seeing or what your colleagues are seeing as well. (GP9)

Fears around missing a sick child characterised the clinical context, with the 'anticipated regret' cognitive bias evident throughout interviews. Clinicians consistently expressed fear of the consequences of missing a sick child, which affected their decision-making. Fears centred on children's deteriorating health and medicolegal concerns.

Do you let the kid go home with the fever etcetera, or do you start antibiotics and go home and relax [laughs]. (GP21)

I think it's about doctors feeling scared of missing significant illness and kids getting sick and [clinicians] getting sued. (GP3)

### 'I rarely prescribe antibiotics': self-presentation as not overprescribing

Clinicians tended not to perceive themselves as (or present themselves to be) overprescribers, showing awareness of antimicrobial resistance and the ineffectiveness of antibiotics for viral infections. Means of distancing themselves from the problem included clinicians describing themselves as distinct from other clinicians or from historical medical practice. This perception suggested no need for current practice to change.

I think antibiotics are overused (–) and I do think the nurses are much better at making the decision not to use them than some of the GPs. (NP16)

**Table 3** Summary of topics explored and themes identified

| Broad category of questions | Topic explored | Inductive themes identified |
|---|---|---|
| (1) Exploration of existing practice in current clinical context | General context | **Fear, risk and safety (children as a vulnerable group)** |
| | RTI diagnostic decision-making and management | Role of GP<br>Uncertainty<br>Dual system decision-making<br>**Probabilistic reasoning—likely to be a virus**<br>**Cognitive bias in decision-making**<br>Parent factors in management choices: worry, expectations, competence<br>**Self-presentation as not overprescribing** |
| | Infection surveillance in the current context | Anecdotal or no evidence gathered |
| | What is needed by clinicians to help with uncertainty | No clear need identified |
| (2) Response to intervention materials (as presented) | Perceived impact of the intervention | **Management decisions do not need surveillance information (all known)**<br>Impact unknown<br>POSITIVE:<br>**Supporting decision-making post hoc**<br>Clinician confidence in viral diagnosis<br>Cognitive bias effects<br>Reducing reconsultation rates<br>Reducing antibiotic prescribing<br>► Other potential positive effects<br>NEGATIVE:<br>**Increased risk (missing the sick child)**<br>Adding complexity<br>Accuracy and representativeness of content |
| | What do clinicians want from the intervention | CONTENT:<br>**Clinician preferences for threat information**<br>Symptom duration<br>DELIVERY:<br>► Accessibility<br>► Recipient—clinician, nurse or practice manager<br>► Shared use with patients in the consultation |
| | Barriers | Information overload<br>Lack of time<br>Lack of fit with clinician role |
| | Perceived utility—will the clinician use it? (implementation) | In an ideal world |

Bold text denotes key themes, presented in detail.
GP, general practitioner; RTI, respiratory tract infection.

I think I rarely prescribe antibiotics. (GP19)

I think we are all aware that maybe people in the past received antibiotics that maybe wouldn't have helped, and actually causes problems later on through resistance and so on, or even side-effects to the patients, so we're aware that most of these things are due to viral illnesses. (GP4)

I have relatively high threshold for antibiotic use. Probably higher than some of my more experienced colleagues I would say. (GP5)

Most clinicians interviewed spoke of offering patient reassurance around viral illness, and there was a strong emphasis on the importance of safety netting and returning for repeat consultations, educating parents about what risks to look out for. Some clinicians talked in this context of normalising infections, as well as empowering parents to manage them, rather than giving antibiotics.

I always do a lot of safety-netting with sick children. I give very clear instructions of what to look out for […] if they're getting worse, I'd prefer them to be seen again, rather than just giving antibiotics. (GP1)

Some clinicians asserted that their default approach is not to prescribe, assuming all paediatric RTIs are viral. By contrast, there was acknowledgement from more than one clinician of a default to prescribing in some circumstances:

> Sometimes, on a busy day, you're tired towards the end of the surgery. You don't want to put up with it. Easy option is to give the antibiotic. (GP6)

### Response to information

Mixed responses were elicited (interestingly both *within* and *across* interviews) to the prototype intervention. Some responses indicated uncertainty of the potential impact on decision-making ('I don't know how much it influences prescribing until I sort of, until it's done' NP14), while some responses recognised potential for increasing confidence in viral diagnosis ('so I can see just reading this top bit about knowing what the local bugs are may help reduce uncertainty about the cause' GP8) or reducing antibiotic prescribing ('if I knew that they were circulating I'd be like, 'oh, okay, they're more viral,' and so possibly less likely to need treating' NP14). There was additional recognition of potential to enhance patient explanation. Many comments were negative, however.

### 'What's it gonna change?': management decisions do not need surveillance information

Clinicians consistently reported that the main information (high viral prevalence and common symptomatic profiles) is known, making the intervention unlikely to impact on clinical decisions. Microbiological information differentiating between viruses did not fit with the clinicians' perceived role of identifying a seriously ill child, or 'bacterial infection', from general self-limiting viruses. In line with clinicians' perceived role as treating the *individual child* (see above), several described the need to fully assess each child, representing clinical judgement as of highest importance. Crucially, many indicated that surveillance data would not change their paediatric RTI management.

> I'm not sure that would make a huge impact on my management, because I don't say to them, 'Oh, I think you've got RSV' [respiratory syncytial virus] or 'I think you've got rhinovirus.' (GP4)

> I guess we know already that that's going to be viral, that's not a sign of a bacterial infection, so it wouldn't….having that confirmed with the results, and seeing that there's a peak at the moment, wouldn't really change my management I don't think. (GP8)

> I mean, to some extent, it's a bit of, well, 'so what?' because none of that is actually going to make any difference to my management. It's really what they're

like clinically, and particularly with viruses. So, I, I'm not sure how it's particularly going to help. (GP13)

> Yeah, fine, what's we gonna do with it? What's it gonna change? (GP3)

### 'It's a bit dangerous': concern it might increase the risk of missing the sick child

In contrast, several clinicians expressed concern about the potential to increase the risk of missing, and therefore not treating, a sick child. This was perceived as jumping to conclusions and reducing the clinician's impetus to make a full clinical assessment of each child as an individual. There was a sense of worry and mistrust of surveillance in this context:

> It's a bit dangerous to start putting stuff down to some other thing that's going round […] So probably I try and ignore data like that […] Yes, I'd still be worried that I wouldn't want to use group data to cloud what the individual was coming in with. (GP10)

> I think it definitely could be helpful, but it could also make you jump to that conclusion rather than fully assessing something, which would make…you don't want to miss something else by just ignoring…that it's gonna be that because that's what's going around. You have to be a little bit careful. (GP1)

Given the 'anchoring and adjustment' bias already outlined, which a population-level surveillance intervention could help clinicians *avoid*, there was interestingly some *concern* about the intervention having a cognitive biasing effect in increasing viral diagnosis, particularly in terms of availability and salience, the very elements the intervention is designed to address:

> Doctors are just normal people as well and we've—our brains work in the same way as a lot of other people's. If you shove something in our face repeatedly we're going to think about that a lot more than all the other less likely causes […] (GP7)

> Yes. (Int)

> Sometimes we can be suggestable as well. (GP7)

This further indicates clinicians' sense of the importance of clinical judgement over population data.

Three clinicians expressed concerns about the accuracy and relevance of surveillance content, indicating the importance of information source credibility. One questioned the representativeness compared with the patient population, characterising families who contribute to public health surveillance as different from hard-to-reach patients who 'come up a lot and get ill' (GP3). Others expressed that their patient population was different

from the wider population (eg, different ethnic mix), indicating potential reasons not to trust population data above clinical assessment. Perhaps there was a sense of feeling safer to do things the way they normally do.

### 'Being more sure': supporting decision-making post hoc
A strong theme came through of the intervention increasing clinicians' confidence by supporting decision-making post hoc, boosting their sense of the accuracy of a viral diagnosis *already* made, rather than contributing to the decision-making process. This was mainly reflected in the context of enhancing patient explanation, providing reassurance, credibility and trust in the clinician's decision.

> If I do decide then it is a bug, and I know that there is one going around, that would be really helpful to be able to say that [to the carer/patient]. (GP10)

> Yeah, just to relay that information with more reassurance, saying that, 'Yes, this is what…the likely cause of the symptoms.' Being more sure about that and relaying that information to parents. (GP6)

> Often it's helpful to show something tangible like a graph or a picture and that sort of validates what they're telling you, and what you're telling them. (GP7)

This potentially reflects clinicians' perceptions as appropriate rather than overprescribers of antibiotics, with no change to diagnostic/management decision-making required. In these examples, by contrast to intervention intentions, enhanced confidence related to *explanation* rather than *diagnosis*.

### What do clinicians want from a surveillance intervention?
#### Content preferences
Typical symptoms duration was consistently identified as useful content ('the symptoms durations data there is actually is incredibly helpful…because I think maybe our perceptions and also parents' perception is that it should be a lot shorter than that' GP12). Although the information presented was based on existing published research, clinicians expressed surprise at higher than expected duration, recognising this as important information to impart to carers/patients.

#### 'Be more aware of the risk': clinician preferences for threat information
The main information clinicians reported wanting from the intervention were: new threats to child health; unusual symptoms presenting within viral patterns; and what to do differently from usual. Information on regular circulating minor viruses was seen by many as of limited interest, perhaps related to their role as assessing the danger to the child and not perceiving themselves as overprescribers. Desire for management or safety-netting

information was expressed, wanting the intervention to incorporate concerning (instead of reassuring) elements.

> I think it would be useful to know if, for example, the RSV was leading to more admissions and children were more unwell with the RSVs compared to influenza or the Rhinovirus […] I suppose that could heighten your awareness of if you get these symptoms they need to be more aware of the risk or look more carefully at the child possibly. (GP9)

> I'd want to know something that would make an impact on the advice that you're giving parents and also for us to not be so reassured when we eyeball a child that it… 'oh no, this is just more of a common cold' […] if there are ones that are a bit out of the blue and worrying, even if there are fewer cases of them, if they're potentially going to have more of a devastating impact on children, you've got a bit of a heads up about that. (GP8)

#### Delivery preferences
In terms of intervention delivery, clinicians expressed the (expected) barriers of information overload and time pressure, asserting that information of this kind must be easily accessible ('one click' on a computer), with some expressing concern that the intervention itself could add complexity and contribute to information overload.

There were mixed ideas concerning the best recipient of surveillance information, whether GPs, NPs or the practice manager to then disseminate key points to clinicians. This latter delivery fit with clinicians' desire to be alerted to information about changes to the clinical picture or heightened risks, highlighting that the proposed intervention may not fit with clinicians' perceptions of their (risk-oriented) role. Most were positive about sharing surveillance information with carers/patients within consultations.

#### Perceived utility
Overall there was a sense that surveillance information of the kind presented may be used 'in an ideal situation' (GP2), but was not seen as necessary.

> It would help but it's not needed, it's like the cherry on the cake [laughs]. (GP20)

### DISCUSSION
This research provides new evidence regarding clinician decision-making and psychological influences relevant to paediatric RTI treatment and intervention development, and attitudes towards population data.

Confidence in clinical judgement was evident throughout key themes: self-perception as not overprescribing, surveillance information as unnecessary, or to

confirm decisions post hoc, wariness of clouding clinical judgement. Despite this, clinicians presented uncertainty and cognitive biases affecting prescribing decisions. Interviews demonstrated that anticipated regret for missing a sick child characterises both decision-making for paediatric RTIs and responses to a potential intervention, evident in perceiving surveillance information as contributing risk.

The wariness of using population data to inform clinical practice is interesting, particularly in the context of clinicians' current approach; assuming high viral prevalence, knowledge of circulating infections anecdotally informed and diagnosis based on 'assessment of the individual' could be seen as elements of 'representativeness' bias (table 1). Perceiving it as a concern rather than a benefit contrasts recommendations for clinician decision-making (see the Introduction section),[22 23] and opposes intentions behind an intervention designed to reduce perceived risks associated with children's RTI. Clinician preference for health threat-related information also opposes intervention intentions, indicating clinicians' risk-oriented role; a challenge for developing interventions implicitly designed to reassure. Rather than aiming to shift current practice, clinicians' perceived need for change was in response to changed (ie, increased) environmental risks or unusual events. To gain 'buy-in' from clinicians about the need to shift prescribing practices, perceptions may need disrupting.

This study shows more work is needed for infection surveillance to be a useful tool for clinicians. Our experimental test of a prototype intervention including surveillance information with parents (Schneider *et al*, submitted) indicated that other elements may be more influential than surveillance information per se. In our parallel study, parents were positive about clinicians sharing surveillance information in consultations.[33] Clinicians were similarly positive about sharing surveillance information with parents/carers of children with RTI, citing enhanced patient explanation and reassurance. This indicates potential for surveillance data to aid *patient* decision-making (eg, reconsultation). Clinicians identified symptom duration as useful and surprisingly longer than expected (despite being published evidence), a finding echoed by parents.[33] Simply promoting awareness of paediatric RTI symptoms duration may enhance decision-making. Brewer and colleagues found that anticipated regret for *inaction* under conditions of assumed responsibility retains high salience with large impacts on behaviour,[40] which is pertinent to primary care decision-making. Intervention development needs to account for this strong motivator, one example being to harness this to elicit anticipated regret about negative consequences of prescribing unwarranted antibiotics.

Wariness was evident in some interviews about the interviewer assessing clinicians' adherence to prescribing guidelines. There is now wide knowledge of antimicrobial resistance. These factors may contribute to clinicians' presentations as appropriate prescribers, particularly when interviewed about an intervention aiming to reduce antibiotic prescribing.

It may be that infection surveillance is not the best focus for developing an intervention to enhance primary care paediatric RTI management. We must acknowledge that this study elicited perceptions of a prototype intervention, rather than measuring actual intervention impact, which would require a pilot and full trial to assess. This research focused on GP responses, with only three NPs interviewed. In the few practices that had NPs there were fewer NPs than GPs to invite to interview. Data saturation may not have been reached in our NP subsample so findings may be more transferable to GP than NP populations.

This study raises valuable topics to explore in future research: (1) ways to harness the behavioural motivator of anticipated regret (eg, negative consequences of *prescribing* rather than not); (2) examining clinicians' reluctance to trust population data (vs recommendations for epidemiological assessment) to aid probabilistic clinical reasoning; (3) how to present reassuring information to a risk-oriented group; and (4) careful consideration of the potential need to disrupt confidence in clinical judgement in order to modify clinician prescribing. Perhaps future work could go beyond the 'so what' to a sense that something's 'gonna change'.

**Author affiliations**
[1]Centre for Academic Child Health, Bristol Medical School: Population Health Sciences, University of Bristol School of Social and Community Medicine, Bristol, UK
[2]NIHR Collaboration for Leadership in Applied Health Research and Care West and NIHR Health Protection Research Unit in Evaluation of Interventions, University of Bristol School of Social and Community Medicine, Bristol, UK
[3]Centre for Academic Primary Care, NIHR School for Primary Care Research, University of Bristol School of Social and Community Medicine, Bristol, UK
[4]NIHR Health Protection Research Unit in Evaluation of Interventions, Centre for Academic Primary Care, University of Bristol School of Social and Community Medicine, Bristol, UK
[5]Department of Health and Social Sciences, University of the West of England, Bristol, UK
[6]Centre for Academic Primary Care, University of Bristol School of Social and Community Medicine, Bristol, UK

**Contributors** ECA was responsible for developing the current research question and secured funding and ethical approval, coordinated the study, led on data collection (interviews) and full analysis, wrote the paper and will act as guarantor. ADH was responsible for the idea of developing a paediatric infection surveillance intervention. IL developed mock-up infection surveillance materials to present within interviews and helped with data collection and analysis. JMK, TM and CC advised on qualitative methods and contributed to analysis. All authors have read, commented on and approved the final manuscript.

**Funding** The study is supported by the National Institute for Health Research Health Protection Research Unit (NIHR HPRU) in Evaluation of Interventions at the University of Bristol, in partnership with Public Health England (PHE). ADH was funded by NIHR Research Professorship (NIHR-RP-02-12-012). JMK was partly funded by NIHR HPRU in Evaluation of Interventions and NIHR Collaboration for Leadership in Applied Health Research and Care West (NIHR CLAHRC West) at University Hospitals Bristol NHS Foundation Trust (UHBT).

**Disclaimer** The views expressed are those of the authors and not necessarily those of the NHS, the NIHR or the Department of Health and Social Care.

**Competing interests** None declared.

**Patient consent for publication** Not required.

**Ethics approval** The South-West Frenchay Bristol Research Ethics Committee approved the study (reference: 15/SW/0264).

**Provenance and peer review** Not commissioned; externally peer reviewed.

**Author note** The work was mainly completed within the Centre for Academic Primary Care, Population Health Sciences, University of Bristol. The University of Bristol acted as study sponsor. Doctoral supervision was provided for the lead author (ECA) by TM within Health and Social Sciences, University of the West of England (UWE). ECA submitted a body of work from which this paper was drawn as partial submission for the award of Professional Doctorate in Health Psychology.

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
