## [Reviewer comments · BMJ Paediatrics Open]

ARTICLE DETAILS

TITLE (PROVISIONAL)	Primary care clinicians' views of paediatric respiratory infection surveillance information to inform clinical decision-making: a qualitative study
AUTHORS	Anderson, Emma; Kesten, Joanna; Lane, Isabel; Hay, Alastair; Moss, Timothy; Cabral, Christie

VERSION 1 – REVIEW

REVIEWER	Reviewer name: Tim Usherwood Institution and Country: The University of Sydney, Australia Competing interests: None
REVIEW RETURNED	01-Jan-2019

GENERAL COMMENTS	The reason for undertaking this study is unclear. The authors do not provide a credible hypothesis for why provision of near-real time information about locally circulating respiratory viruses might reduce antibiotic prescribing for respiratory infections. The only empirical evidence they provide in support of this putative effect is a systematic review (reference 42). However, of the three studies in that review which reported on antibacterial prescribing: •In the first (uncontrolled) study, provision of viral surveillance data was confounded by an educational program•The second study was purely observational and included no intervention by the investigators•The third, a cluster RCT, is unpublished but included information about GAS prevalence in addition to that of viruses. Of note, many of the study participants also expressed the view that surveillance data would not change their management of patients (Page 10). Methodologically, the study was well conducted although data saturation may not have been achieved for NPs, and analysis did not explore possible differences between GPs and NPs in detail. The authors mention these points in the Discussion. The authors situate their work in the context of research into cognitive biases in medical decision making, and discuss their findings critically within that context. Their study may provide useful evidence to inform future development of interventions to support appropriate prescribing for childhood respiratory infections. However, as the authors note in the Discussion, "It may... be that infection surveillance is not the most appropriate focus for developing a successful intervention".. There are some additional issues that the authors should address: •Reference 6 was published over 10 years ago, and does not support the statement "RTIs are the most common problem managed by primary care". This is not the case in Australia, for example. Do the authors mean "in children"?
--

	 •References 19 & 21 are identical •The last column of Table 2 is “Years practicing as a GP” but the table includes NPs •Despite its title, Table 4 does not provide ‘principles’ but is a list of observations, suggestions and comments.
--	---

REVIEWER	Reviewer name: Malcolm Brodrie Institution and Country: Newcastle University/Great North Children's Hospital, UK Competing interests: None
REVIEW RETURNED	05-Feb-2019

GENERAL COMMENTS	I enjoyed reading this interesting manuscript. It is well-written and easy to follow. The manuscript reports part of a larger study and specifically relates to some qualitative work examining the opinions of primary care clinicians around a hypothetical surveillance system to provide them with up to date local data re. the epidemiology and aetiology of respiratory tract infections in children to then aid with clinical decision-making, mainly antibiotic prescribing. The responses and themes are presented and discussed nicely in my opinion. Most would agree I think that many of these themes are what one would expect primary care clinicians concerns and opinions to be. Therefore the points below are relatively minor points/comments only:  - A crucial point seems to be that each child is obviously an individual and each case must warrant thorough assessment irrespective of the current prevalence of a particular respiratory virus - I do not regard that point of view as defensive, rather just good clinical practice. - I am surprised that NICE Sepsis guidance did not crop up in the interviews - this has impacted on hospital paediatric practice certainly. - Also surprised that 'delayed prescriptions' or the quality and robustness of safety-netting advice did not crop up either - I know that this was a hypothetical intervention and therefore it is not fair to question it really, but logistically how would high quality, up to date, local aetiological data ever be generated in reality? - Those with an interest in respiratory infection in children would likely point towards increasing evidence of polymicrobial infection in children who become more symptomatic and require hospitalisation for example - if one works backwards with the benefit of hindsight the history will normally start with 'viral' type symptoms in most cases - a key challenge it seems is to be able to prospectively identify these children - Accurate point of care testing is mentioned briefly, I agree potentially very useful in this context - Ref 28 the journal isn't included, is an important reference
--

VERSION 1 – AUTHOR RESPONSE

Reviewer: 1

Revision # Comments to the Author

1 The reason for undertaking this study is unclear. The authors do not provide a credible hypothesis for why provision of near-real time information about locally circulating respiratory viruses might reduce antibiotic prescribing for respiratory infections. The only empirical evidence they provide in support of this putative effect is a systematic review (reference 42). However, of the three studies in that review which reported on antibacterial prescribing:

- In the first (uncontrolled) study, provision of viral surveillance data was confounded by an educational program
- The second study was purely observational and included no intervention by the investigators
- The third, a cluster RCT, is unpublished but included information about GAS prevalence in addition to that of viruses.

Of note, many of the study participants also expressed the view that surveillance data would not change their management of patients (Page 10).

Author response:

Thank you for this comment. We have considerably restructured the introduction which hopefully makes the rationale clearer – i.e. our hypothesis is that surveillance data may reduce clinical uncertainty, a driver of antibiotic prescribing, as well as boosting deductive reasoning (away from cognitive bias). We are aware that the evidence in the systematic review was not robust and state this, adding that more research is needed. The first step before developing and testing a surveillance intervention is to explore stakeholders' views, which this research was designed to do.

Revision made:

Introduction restructured and refined including moving the systematic review reference to here.

2 Methodologically, the study was well conducted although data saturation may not have been achieved for NPs, and analysis did not explore possible differences between GPs and NPs in detail. The authors mention these points in the Discussion.

The authors situate their work in the context of research into cognitive biases in medical decision making, and discuss their findings critically within that context. Their study may provide useful evidence to inform future development of interventions to support appropriate prescribing for childhood respiratory infections. However, as the authors note in the Discussion, "It may... be that infection surveillance is not the most appropriate focus for developing a successful intervention" ..

Author response:

Thank you for these comments.

Revision made:

n/a

There are some additional issues that the authors should address:

3 •Reference 6 was published over 10 years ago, and does not support the statement “RTIs are the most common problem managed by primary care”. This is not the case in Australia, for example. Do the authors mean “in children”?

Author response:

Thank you for your query. Yes we do mean in children. We have adjusted the text to make this clearer.

Revision made:

Introduction, sentence two:

“RTIs are the most commonly managed problem of childhood managed in primary care.”

4 •References 19 & 21 are identical

Author response:

Thank you for pointing this out. We have amended the references.

Revision made:

Reference list – 19 is now the only reference for the Horwood paper

5 •The last column of Table 2 is “Years practicing as a GP” but the table includes NPs

Author response:

Thank you for pointing this out. We have revised the column heading.

Revision made:

Table 2, final column heading

6 •Despite its title, Table 4 does not provide ‘principles’ but is a list of observations, suggestions and comments

Author response:

In cutting the discussion down by half, we deleted Table 4 altogether

Revision made:

Deleted table 4

Reviewer: 2

Revision # Comments to the Author

7 I enjoyed reading this interesting manuscript.

It is well-written and easy to follow.

The manuscript reports part of a larger study and specifically relates to some qualitative work examining the opinions of primary care clinicians around a hypothetical surveillance system to provide them with up to date local data re. the epidemiology and aetiology of respiratory tract infections in children to then aid with clinical decision-making, mainly antibiotic prescribing.

The responses and themes are presented and discussed nicely in my opinion.

Most would agree I think that many of these themes are what one would expect primary care clinicians concerns and opinions to be. Therefore the points below are relatively minor points/comments only

- A crucial point seems to be that each child is obviously an individual and each case must warrant thorough assessment irrespective of the current prevalence of a particular respiratory virus - I do not regard that point of view as defensive, rather just good clinical practice.

Author response:

Thank you for this point. We agree that assessment of each child as an individual is clinically necessary. What was interesting in our results was that in their individual assessments, clinicians' decision-making appeared to be influenced by cognitive biases. As described in the introduction, guidelines advise incorporating epidemiological evidence into diagnostic assessment – whereas clinicians in this study were wary of using population data at all (rather than welcoming it as a tool to aid their diagnostic reasoning). Viewing the provision of population data as bringing potential dangers of clouding rather than aiding clinical reasoning was particularly interesting - especially when we know that over-prescription of antibiotics is a population health concern.

Perhaps suggesting 'defensiveness' about clinical judgement gave the wrong tone. We have deleted this.

Revision made:

Deleted mention of clinician 'defensiveness' in discussion (now discussion para 2).

8 - I am surprised that NICE Sepsis guidance did not crop up in the interviews - this has impacted on hospital paediatric practice certainly.

Author response:

Yes, interesting that our clinicians did not mention this.

Revision made:

n/a

9 - Also surprised that 'delayed prescriptions' or the quality and robustness of safety-netting advice did not crop up either

Author response:

This paper presents the main findings that we felt were relevant to the potential intervention (trying to avoid replicating too much known information). In this vein we chose to focus the existing practice section on diagnostic decision-making and antibiotic prescribing. While there was brief mention of delayed prescriptions in interviews, it was not presented as a usual course of action.

Safety netting was described by the clinicians we interviewed, which in fact related to their statements about not prescribing antibiotics.

We have added a paragraph and quotation to illustrate this.

Revision made:

- Existing practice section of results: "I rarely prescribe antibiotics": Self-presentation as not over-prescribing: Added Paragraph 2 and GP1 quotation

10 - I know that this was a hypothetical intervention and therefore it is not fair to question it really, but logistically how would high quality, up to date, local aetiological data ever be generated in reality?

Author response:

Yes, a good question. The wider study in which this qualitative study was embedded went a long way towards answering.

We state in the introduction (now para 4):

The Evaluation of Enhanced Paediatric Respiratory Infection Surveillance (EEPRIS) Study²⁵, within which this qualitative study nested, demonstrated feasibility and acceptability of community surveillance (parents prospectively gathering microbiological and syndromic data) of paediatric RTI²⁶.

We have been asked to reduce the word count considerably

Revision made:

Introduction, para 4:

Microbiological and syndromic incidence data are routinely collected (e.g. by Public Health England and QRESEARCH). We have demonstrated the feasibility of primary care²⁶ and parent collected²⁷ microbial sampling that could also be used for surveillance, the being the Evaluation of Enhanced Paediatric Respiratory Infection Surveillance (EEPRIS) Study²⁸, within which this qualitative study was nested.

11 - Those with an interest in respiratory infection in children would likely point towards increasing evidence of polymicrobial infection in children who become more symptomatic and require hospitalisation for example - if one works backwards with the benefit of hindsight the history will normally start with 'viral' type symptoms in most cases - a key challenge it seems is to be able to prospectively identify these children

Author response:

This is an interesting point. We have, however, been asked to reduce the word count considerably so are restricting our paper to the interview results and this intervention.

Revision made:

n/a

12 - Accurate point of care testing is mentioned briefly, I agree potentially very useful in this

Context

Author response:

Thank you for this comment.

Revision made:

n/a

13 - Ref 28 the journal isn't included, is an important reference

Author response:

Thank you for noticing this. We have revised the reference

Revision made:

Updated Anderson et al 2018 reference #27